# Leverage Effective Deep Learning Searching Method for Forensic Age Estimation

**DOI:** 10.3390/bioengineering11070674

**Published:** 2024-07-02

**Authors:** Zhi-Yong Zhang, Chun-Xia Yan, Qiao-Mei Min, Yu-Xiang Zhang, Wen-Fan Jing, Wen-Xuan Hou, Ke-Yang Pan

**Affiliations:** 1Key Laboratory of Shaanxi Province for Craniofacial Precision Medicine Research, College of Stomatology, Xi’an Jiaotong University, Xi’an 710004, China; zzy20011126@mail.xjtu.edu.cn; 2College of Forensic Medicine, Xi’an Jiaotong University Health Science Center, Xi’an 710061, China; 3Department of Orthodontics, The Affiliated Stomatological Hospital of Xi’an Jiaotong University Health Science Center, Xi’an 710004, China; 4Department of Radiology, The Affiliated Stomatological Hospital of Xi’an Jiaotong University Health Science Center, Xi’an 710004, China; 5School of Electronic and Information Engineering, Xi’an Jiaotong University, Xi’an 710049, China

**Keywords:** age estimation, deep neural network (DNN), orthopantomograms (OPGs), Neural Architecture Search (NAS)

## Abstract

Dental age estimation is extensively employed in forensic medicine practice. However, the accuracy of conventional methods fails to satisfy the need for precision, particularly when estimating the age of adults. Herein, we propose an approach for age estimation utilizing orthopantomograms (OPGs). We propose a new dental dataset comprising OPGs of 27,957 individuals (16,383 females and 11,574 males), covering an age range from newborn to 93 years. The age annotations were meticulously verified using ID card details. Considering the distinct nature of dental data, we analyzed various neural network components to accurately estimate age, such as optimal network depth, convolution kernel size, multi-branch architecture, and early layer feature reuse. Building upon the exploration of distinctive characteristics, we further employed the widely recognized method to identify models for dental age prediction. Consequently, we discovered two sets of models: one exhibiting superior performance, and the other being lightweight. The proposed approaches, namely AGENet and AGE-SPOS, demonstrated remarkable superiority and effectiveness in our experimental results. The proposed models, AGENet and AGE-SPOS, showed exceptional effectiveness in our experiments. AGENet outperformed other CNN models significantly by achieving outstanding results. Compared to Inception-v4, with the mean absolute error (MAE) of 1.70 and 20.46 B FLOPs, our AGENet reduced the FLOPs by 2.7×. The lightweight model, AGE-SPOS, achieved an MAE of 1.80 years with only 0.95 B FLOPs, surpassing MobileNetV2 by 0.18 years while utilizing fewer computational operations. In summary, we employed an effective DNN searching method for forensic age estimation, and our methodology and findings hold significant implications for age estimation with oral imaging.

## 1. Introduction

Identification of victims and suspects is crucial in a variety of scenarios, such as mass disasters, as well as criminal and civil cases [1,2,3]. The teeth play a key role in this process as they are incredibly durable and able to withstand a wide range of extreme conditions [4,5]. This resilience makes them invaluable for accurately identifying individuals in challenging circumstances. Each tooth has unique characteristics such as shape, coloration, wear pattern, pathology, and position. These characteristics differ from person to person and are important for identification. The arrangement of teeth in the oral cavity is also unique to each individual, making them important for identification purposes. Dental characteristics are personalized traits that can help in identifying individuals [6].

Age serves as a vital aspect of an individual’s identity, acting as a crucial verification in forensic procedures. The accurate estimation of age for person holds paramount importance in forensic investigations. Age estimation relies on the physiological changes that occur during the aging process within the human body. Teeth and bones are widely utilized anatomical structures for predicting age accurately. Skeletal maturation and degeneration give rise to morphological alterations that can be used to estimate age reliably. Evaluating the size, shape, and degree epiphyseal ossification in bones allows for age determination. Various skeletal elements, for example, the clavicle [7], ilium, pubis [8], knee [9,10], and foot [11,12], are extensively utilized in the study of bone age estimation, with a special focus on the hand-wrist region [13,14,15]. Conversely, dental age estimation is viable method employed in forensic practice.

The estimation of dental age relies heavily on aging changes and can be classified into three main groups: histological, formative and degenerative changes. Formative changes involve the finalization of formation, the emergence of the crown, and the growth of the roots. Degenerative changes consist of root resorption, abrasion, secondary dentin deposition, root transparency, and deposition of cementum. These formative characteristics, along with degenerative and histological markers, undergo alterations throughout the processes of human growth and development. Consequently, these characteristics can be utilized for dental age estimation. Previous studies have employed to predict age; but these require tooth sections and are not applicable in vivo. In cooperation to such approaches, radiographic methods offer simplicity, reproducibility and non-invasiveness, making them indispensable tools for accurate age estimation [16].

Dental age is commonly determined by forensic professionals through the analysis of developmental stages [17,18] or by assessing the level of apical closure [19]. Age estimation can be achieved using digital panoramic radiographs, intraoral periapical radiographs, and computer tomography images. However, it is important to note that radiological analysis is inherently subjective, and many of the features employed are primarily applicable to pediatric and adolescent populations.

In recent years, DNN has been widely used in objective age assessment. This technological innovation has significantly improved the processing capacity and accuracy of current image applications. The technology architecture based on DNN has greatly advanced the field of image analysis through its strong characteristic learning and representation capabilities. The application of the DNN architecture to estimate the age according to OPGs in this particular area makes end-to-end age estimation possible and is expected to maximize the potential of the existing DNN architecture. Especially in the research of dental age estimation, deep learning techniques have become the preferred analytical method due to their accuracy and speed advantages. Compared to traditional methods, deep learning not only enhances the precision of dental age estimation but also significantly reduces the human resources required for forensic investigations. Currently, researchers widely utilize neural networks for age estimation analysis. Some studies [20] have shown that deep learning methods outperform traditional methods in handling complex image recognition tasks, particularly when dealing with different types of X-ray images such as OPG, MRI, and CBCT, where deep learning demonstrates stronger adaptability. Moreover, Barkmn [21] explored the impact of different super-parameters, model complexity, and dataset size on model performance. They used the EfficientNet-B4, DenseNet-201, and MobileNet V3 models to cross-authenticate OPG datasets and change batch size and sample volume. The results showed that EfficientNet-B4 performed best at the batch size of 160, while MobileNetV3 peaked at the batch size of 160. In addition, as the sample volume decreased, the model performance remained relatively excellent, highlighting the importance of super-parameter optimization. The above studies provide a point to consider elements for model selection and optimization, but the model design of neural network often relies on man-made experience and expertise and requires considerable time and effort to test and optimize. There is also a lack of an effective DNN search method for forensic age estimation for a neural network architecture that optimizes specific tasks.

In this research, a dataset of dental radiographic images from a specialized stomatological hospital was compiled, featuring labeled OPGs of 27,957 subjects. The age range of the subjects in the dataset spans from 1 to 93 years old, and the median age is 27. The accuracy of the age annotations was verified by cross-referencing with their corresponding ID card. It is worth mentioning that the dental dataset displays unique features.

Moreover, our study focuses on leveraging DNN architectures for end-to-end age estimation using OPGs. To maximize the potential of existing DNN architectures, we thoroughly analyzed various characteristics of these models for accurate age estimation in dental applications. Specifically, we investigated the effectiveness of elements, DNN depth, multi-branch architectures, and convolutional kernel size.

We utilized a Neural Architecture Search (NAS) approach to investigate appropriate deep neural network (DNN) structures for predicting age. Inspired by the characteristics of existing DNN models, we aimed to discover efficient DNN structure that offer optimal accuracy and performance. Our search space design primarily encompasses four aspects: (1) multi-branch architectures; (2) asymmetric convolution kernels; (3) network depth variations and (4) early layer feature reuse. Ultimately, our NAS-derived DNN architectures exhibited superior performance compared with the baseline models.

## 2. Methods

### 2.1. Exploration of Neural Network Elements for OPG

An X-ray image exhibits distinct characteristics compared to traditional RGB images. To date, there is a lack of specific studies on the design of DNN models for X-ray images. Although all images share similar texture, they do not possess irregular patterns. To investigate the efficacy of different DNN architecture for age estimation, we initially tried various existing models: Inception-v [22], incorporating a split-transform-merge concept, which utilizes shortcut structures and reuses previous feature layers; VGG16 [23], with a vertical connection structure; MobileNetV2 [24], a lightweight network with a structure; ShuffleNetV2 [25], employing shuffle operations; and EfficientNet [26], discovered through NAS. These models comprehensively encompass the existing rules of neural network design. Additionally, we conducted a comparative analysis on the X-ray image dataset, evaluating ResNet [27], ResNeXt [28], and DenseNet [29] with different depths of network architecture to investigate the influence of DNN depth on forecast accuracy.

The estimation of age for all the previously mentioned DNN models was calculated using mathematical expectations. Inspired by DEX [30], we firstly established the age probability distribution for every image. Then, we used mathematical expectations to generate the final age estimation, as shown in Equation (1):(1)$$y=\sum_{k}p_{k}\cdot l_{k}$$
where the *p_k_* is the estimation probability that the input image belongs to label *l_k__,_* the value of label is the real age of all the dental image. The *k* is the index of age for all the dental image, it ranges from 0~75.

According to research on characteristic DNN, four rules have been identified for designing effective age estimation network models for X-ray images. Firstly, it is important to incorporate multi-branch architectures, such as the split-transform-merge structure, to efficiently extract features from X-ray images. Additionally, the utilization of asymmetric convolution kernels, such as 7 × 1, 1 × 7, 3 × 1 and 1 × 3 convolutions, plays a crucial role in successful feature extraction. Moreover, determining the optimal depth for DNN architectures is critical in achieving a balance between comprehensive feature extraction and preventing overfitting. Lastly, it was determined that reusing early layer features in the DNN architecture may actually hinder feature extraction. These guidelines can help improve the effectiveness of age estimation models for X-ray images, ultimately leading to more accurate and reliable results in this field.

Rule 1 delves into the complex structure of multiple branches: architectures with multiple branches, including inception design and group convolution, have proven to be effective in extracting features, as seen in the achievements of models like ResNeXt and Inceptionv4. Our research underscores the importance of multi-branch architecture for OPG datasets. X-ray images use each individual tooth as a basic component, making them suitable for feature extraction using multi-branch structures. The use of various convolutional or pooling operations in a single layer enables efficient feature extraction and minimizes redundancy in the convolutional layer.

Rule 2 delves into the consideration of which convolution kernels are most suitable for inclusion in the DNN for analyzing X-ray images. Due to the unique attributes of X-ray images, the distribution of features can differ greatly in terms of length and width locations. Thus, the utilization of asymmetric convolutions, such as the 1 × 7 convolution, proves to be effective in accurately extracting dental features.

Rule 3 delves into the connection between network depth and the resolution of OPG. This exploration is particularly important when dealing with OPGs, as they are typically less complex than natural images. It is vital to find the right balance in terms of network depth to avoid the risk of overfitting the model. To achieve this, we employed a DNN search methodology to determine the appropriate model depth while keeping the width of the DNN architecture fixed for OPGs.

Rule 4 explores the reuse of features in the initial layers of DNN architectures. Analysis of the image patterns in the OPGs reveals a notable similarity, suggesting that the low-level features in these images also exhibit resemblances. Therefore, repurposing early layer features could potentially introduce more noise.

According to the four rules, we propose a novel DNN model that is more effective for the OPGs dataset. Subsequently, in the next section, the NAS technique is utilized to automatically generate efficient DNN architectures for OPGs.

### 2.2. Model Search Based on Proposed Rules for OPGs

A framework is introduced in this study to effectively search a deep neural DNN architecture for object proposal generation, as illustrated in Figure 1. The framework consists of four main steps: (1) designing the search space, (2) exploring novel architectures, (3) pre-training the model on the ImageNet dataset, and (4) fine-tuning the pre-trained OPG datasets. The four proposed design rules were utilized to effectively design a search space for the purpose of obtaining an architecture that is capable of achieving optimal performance. Subsequently, the network underwent pre-training on the ImageNet dataset. Finally, the DNN architecture was retrained using the comprehensive training dataset of OPGs, leveraging the weights and parameters obtained from pre-training. Following these sequential steps, we successfully derived the final DNN architecture specifically tailored for dental X-ray datasets, enabling accurate age estimation on inference images as depicted in Figure 2.

The exploration of model architecture is crucial in obtaining an appropriate deep neural network (DNN) model for aging estimation on OPGs. Our choice of utilizing PC-DARTS [31] for our search framework was based on its high efficiency in searching and memory utilization. This framework is capable of exploring a multi-branch structure, aligning with the proposed rule 1. PC-DARTS builds upon the renowned differentiable NAS pipeline DARTS [32]. Stacked cells make up the search network, with each cell being a directed acyclic graph (DAG) in which nodes represent feature maps and edges represent mixed operations. With $x_{i}$ representing the ith node, and E$\left(i,j\right)$ representing the edge between $x_{i}$ and $x_{i}$ (i < j), the feature transformation in E$\left(i,j\right)$ can be obtained, represented by $f_{i,j}$, as shown in Equation (2):(2)$$f_{i,j}\left(x_{i}\right)=\sum_{o^{\prime }\in O}\frac{\exp\left(\alpha_{o^{\prime }}^{\left(i,j\right)}\right)}{\sum_{o^{\prime }\in O}\exp\left(\alpha_{o^{\prime }}^{\left(i,j\right)}\right)}o(x_{i})$$

In the operation space O, every element represents a candidate function *O*(·), and the architecture parameters $\alpha_{o}^{\left(i,j\right)}$ must be optimized. During the search process, the network weights w and architecture weights α are adjusted alternately through gradient descent. Once fully optimized, the operation associated with the architecture weight is selected directly.

One drawback of DARTS is its inefficiency in memory utilization, due to the inability of the DARTS framework to facilitate direct search within our OPGs dataset. In order to tackle this problem, PC-DARTS utilizes partial connections between channels, mixing operations for some portions of the edge while keeping others unchanged as identity mappings. Additionally, a channel-sampling mask $S_{i,j}$ is defined to selectively choose or mask specific channels.
(3)$$f_{i,j}^{PC}(x_{i};S_{i,j})=\sum_{o\in O}\frac{\exp\left(\alpha_{o}^{\left(i,j\right)}\right)}{\sum_{o^{\prime }\in O}\exp\left(\alpha_{o^{\prime }}^{\left(i,j\right)}\right)}o\left(S_{i,j}*x_{i}\right)+\left(1-S_{i,j}\right)*x_{i}$$

Partial channel connection in the searching pipeline can lead to instability. In order to address this issue, edge normalization with additional architecture weights, denoted as β, is employed to compute $x_{i}$:(4)$$x_{j}=\sum_{i<j}\frac{\exp\left(\beta^{\left(i,j\right)}\right)}{\sum_{i^{\prime }<j}\exp\left(\beta^{\left(i^{\prime },j\right)}\right)}*f_{i,j}^{PC}(x_{i};S_{i,j})$$

The search is conducted directly on the OPGs dataset rather than using a proxy dataset like CIFAR-10 or ImageNet, taking into consideration the variations in feature distribution among different datasets. Furthermore, we expanded our search scope to include asymmetric convolution and group convolution in accordance with Rule 2. The search scope O encompassed a total of 11 operators:

3 × 3 standard conv

3 × 3 separable conv

5 × 5 separable conv

3 × 3 group conv (groups = 9)

5 × 5 group conv (groups = 9)

1 × 3 then 3 × 1 group conv (groups = 9)

1 × 7 then 7 × 1 group conv (groups = 9)

3 × 3 max pooling

3 × 3 average pooling

identity

zero (no connection)

Each group convolution is followed by a pointwise convolution, similar to the concept of separable convolutions. The technique introduced in PC-DARTS involves utilizing every separable convolution twice within the network structure. The network consists of 8 cells, with 5 regular cells and 3 reduction cells. Specifically, the 3 reduction cells were positioned at the 1st, 3rd, and 6th out of the total 8 positions. The specifics of the explored network will be discussed in the following section.

The cellular topology in the PC-DARTS framework exhibits complexity, rendering it unsuitable for resource-constrained scenarios such as deployment on hardware platforms. Therefore, in this article, we additionally explore lightweight models based on the SPOS [33] to strike a balance between accuracy and inference speed.

The initial search space in the SPOS framework is constructed based on ShuffleNetv2; all candidate blocks in this original search space incorporate a feature reuse structure that conflicts with Rule 4. To address this issue and align with the requirements of Nature journal [34], we have redesigned space using MobileNetV2, another lightweight network. The original MobileNetV2 block lacks the inclusion of asymmetric convolutions; hence, we introduced a parallel asymmetric convolution block as a potential alternative. The configuration of this parallel asymmetrical convolutional unit is depicted on the right-hand side in Figure 3. It presents two key benefits:
(1)The depth of the parallel asymmetric convolution block is equivalent to that of the original MobileNetV2 block, ensuring seamless adherence to Rule 3.(2)Let $C_{in}$ and $C_{out}$ denote the number of input channels and output channels, respectively, in a MobileNetV2 block. Here, EXP2 represents the expand ratios and k represents the kernel size. The resulting output feature map has dimensions h×w (for convenience; we assume that the number of output channels of all layers in the map are *h* × *w*, which does not affect our analysis and conclusion);then, the FLOPs of the block $F_{m}$ is:(5)$$F_{m}=\left({{EXP2\times (C}_{in}}^{2}+C_{in}C_{out}+C_{in}\times k^{2}\right)\times h\times w$$


For our proposed parallel asymmetric convolution block, the FLOPs of the block ${F_{m}}^{\prime }$ is:$${F_{m}}^{\prime }=\left({{EXP2\times (C}_{in}}^{2}+C_{in}C_{out}+C_{in}\times 7\right)\times h\times w$$
(6)$$\frac{{F_{m}}^{\prime }}{F_{m}}=\frac{\left({{EXP2\times (C}_{in}}^{2}+C_{in}C_{out}+C_{in}\times 7)\right)\times h\times w}{\left({{EXP2\times (C}_{in}}^{2}+C_{in}C_{out}+C_{in}\times k^{2})\right)\times h\times w}=\frac{C_{in}+C_{out}+7}{C_{in}+C_{out}+k^{2}}$$

In this paper, we set k=3 or k=5, so $\frac{{F_{m}}^{\prime }}{F_{m}}<1$, and the parallel asymmetric convolution block had less FLOPs than the MobuleNet2 block.

The search space under the SPOS framework is illustrated in Figure 4, where each layer consisted of three candidate blocks. In accordance with Rule 1, every candidate block incorporated at least one depth-wise convolution, resulting in a multi-branch structure. As for Rule 2, the parallel asymmetric convolution block employed both 1 × 7 and 7 × 1 convolutions, which were characterized by their asymmetrical nature. For Rule 3, we set the number of stacking blocks to be 16, with each block having a depth of 3. Considering the presence of an initial layer (the first convolutional layer) and a final classification layer, the network’s depth precisely amounted to 50 layers. Regarding Rule 4, all candidate blocks were reused structures. In general, the search space within the SPOS framework adhered to all four rules. The searched DNN model are shown in Section 4.

## 3. Experiments

### 3.1. Dataset of X-ray

We have already collected a large amount of OPGs, encompassing labeled images, comprising 16,385 females and 11,573 males. The age reached up to 93 years old with a median age of 27 years (Figure 5). Cross-referencing with the information on the ID cards guaranteed the accuracy of the age labels. The majority of images had a resolution of 1536 × 3292 pixels. The age distribution of our age dataset is depicted in Figure 5.

### 3.2. Experimental Settings

In order to obtain high-resolution features for age estimation, all input images were resized to 384 × 384. Image augmentation techniques such as random horizontal flipping, scaling, and translation were applied during the training stage. In the testing phase, both the original test image and its mirrored counterpart were fed into the network, and the average error of the two was calculated as the final error measurement. 

During the searching stage of the PC-DARTS framework, we followed the search settings and hyper-parameters based on the original PC-DARTS model. However, we made two specific adjustments to our approach. Firstly, we partitioned our training dataset into two subsets, with 60% of images used for training and the remaining 40% reserved for architecture hyper-parameters. Secondly, we set the batch size at 16 and the initial learning rate at 0.1. In contrast, for the SPOS framework, we modified the training process by training the super-net for 20,000 mini batches, utilizing the momentum SGD optimizer with a momentum of 0.9 and an initial learning rate of 0.1. Following the inceptionv3 model, we adjusted the batch sizes to 4 × 10^5^ and 64. Additionally, we implemented the label smoothing strategy to optimize the performance of our algorithm. In our study, we applied Label Smoothing Normalization (LSR), which is commonly used in text classification tasks. Like L2 and dropout, it is a regularization method that adds noise to the label to constrain the model. The purpose is to prevent the model from predicting labels too confidently during training, prevent overfitting, and improve the model’s generalization ability.

The networks underwent initial pre-training on the ImageNet dataset and optimization through Adam with a batch size of 16. A momentum value of 0.9 was employed, while the learning rate was initialized to 0.0001, and training continued for 100 epochs. The weight decay was set to 1 × 10^4^ and 4 × 10^5^ in the PC-DARTS framework and the SPOS framework, respectively. All the DNN networks were implemented using PyTorch on a GTX 1080Ti GPU.

### 3.3. Experimental Results for DNN Architecture Searching

In all experiments, we selected mean absolute error (MAE) as the primary evaluation metric. MAE was computed by averaging the absolute differences between predicted ages and actual ages.
(7)$$MAE=\sum_{k=1}^{N}\left|\hat{l_{k}}-l_{k}\right|/N$$
where *l_k_* represents the true age of the test image k, $\hat{l_{k}}$ stands for the predicted age, and *N* denotes the total count of test images.

Figure 6a shows the typical cell and the reduction cell acquired through PC-DARTS, as illustrated in Figure 6b, respectively. During the evaluation stage, a total of eight cells existed, comprising five normal cells and three reduction cells, with each type of cell sharing an identical architecture. We initialized the channel count to 36 for initial model AGENet-Small, and subsequently adjusted it to 54 for our second model AGENet-Large.

The architecture of the searched AGE-SPOS is depicted in Figure 7, where out of the total 16 blocks, half of them (i.e., 8 blocks) were parallel asymmetric convolution blocks, aligning with Rule 1 stated in Section 2.1 This finding further confirms the crucial role played by asymmetric convolution in feature extraction on OPGs datasets.

AGE-SPOS demonstrated superior performance models and exhibited exceptional lightweight characteristics. Specifically, AGE-SPOS (1.0×), out all models with FLOPs, did not exceed 12.59 B (13.25 times the FLOPs of AGENet (1.0×)). Compared to Inception-v4, although the MAE increased by 0.10 years, the FLOPs were reduced by a factor of 21.54. Similar results can be observed when comparing AGE-SPOS with ResNeXt, where the MAE increased by 0.06 years, but the FLOPs decreased by a factor of 13.25. AGENet-Small achieved a reduction in MAE of 0.14 year compared to the lightweight and auto-searched EfficientNet-B0, while our FLOPs were reduced by 22.1%. More detailed results can be found in Table 1 and Figure 8.

### 3.4. Analysis for Our Rules

Based on our comprehensive analysis of the models we have researched and the results obtained, we conducted an in-depth examination of our four rules.

For Rule 1, compared with ResNeXt and Inception-v4, our searched model predominantly employed solutions, further validating this rule.

For Rule 2, in addition to our AGENet and AGE-SPOS, Inception-v4 displayed superior performance and distinguished itself as the sole network incorporating asymmetric convolutions. This assertion is further supported by the configuration of our search cells.

For Rule 3, it is notable that the MAE reached its minimum at a network depth of 50. Subsequently, as the network depth surpassed this threshold, there was a discernible escalation in the MAE. Moreover, when compared to ResNeXt-50, ResNeXt-101 displayed an increase in the MAE of 0.08; similarly, DenseNet-201 showed an increase in the MAE of 0.1 compared to DenseNet-121. The specific network depth was contingent upon the utilized model. Based on our empirical investigations, we recommend a depth ranging from 50 to 90 layers.

For Rule 4, the performance of ShuffleNetV2 was significantly inferior to that of other lightweight models, including ShuffleNetV2, MobileNetV2, EfficientNet and our searched AGE-SPOS. This can be attributed to the reuse of features in ShuffleNetV2. The results obtained from DenseNet further validate this conclusion. Despite its significantly higher FLOPs and parameters compared to lightweight models, DenseNet121 achieved an MAE of only 2.72. It is crucial to note that age estimation using OPGs depends on unique dental characteristics.

In this paper, the Squeeze-Excitation (SE) module was incorporated into AGE-SPOS to investigate its impact on the accuracy of AGE-SPOS. The corresponding results are presented in Table 2. The addition of the SE module to AGE-SPOS (1.5×) demonstrated a further reduction in the network’s MAE by 0.02 years. Incorporating the SE module into AGE-SPOS (1.0×) resulted in an equivalent MAE compared to its pre-SE module counterpart, whereas incorporating the module into AGE-SPOS (0.5×) yielded a significantly higher MAE of 2.61 years, surpassing the original MAE prior to incorporating the SE module. This study aimed to analyze the role of the SE module in weighting different channels within a network. When the number of channels in the network was large (i.e., wide), there tended to be increased redundancy, resulting in certain channels playing a relatively minor role in feature extraction. Currently, employing the SE module to assign weights enhanced outcomes. Nevertheless, in scenarios where the network possessed a limited number of channels (i.e., narrow network), there existed reduced redundancy within the network. At this time, the utilization of SE module for channel weighting had limited impact, leading to decreased network accuracy due to increased training difficulty. Despite its low FLOPs count and minimal computational load on the network, the SE module could not operate in parallel with the convolutional layer during actual reasoning process. After analyzing the output feature map of the preceding layer, the SE module could be inferred based on the aforementioned output feature map. Considering that the primary objective of this research was to identify a network model that effectively incorporates discrimination accuracy and inference efficiency, it is not advisable to incorporate the SE module into the AGE-SPOS explored in this study.

## 4. Discussion

Dental radiographs serve as a fundamental tool in oral clinical practice, playing a pivotal role in the diagnosis of dental diseases. This journal work is an extension version from our conference work, which was published in International Joint Conference on Neural Networks and the conference paper named Exploring Effective DNN Models for Forensic Age Estimation based on Panoramic Radiograph Images [35]. The journal article adds new content for the age estimation compared with conference paper.

For the task of age identification in dental X-ray images, this study builds upon research and employs a research strategy of exploring search. Firstly, it critically analyzes the structure of a network based on four existing hypotheses to assess their validity. Subsequently, these four hypotheses were utilized as prior knowledge to design the search space and explore a lightweight network structure. 

The search space was enhanced by incorporating a module based on parallel asymmetric convolution, which addressed the computational requirements of lightweight networks. When considering the three key indicators of discriminant accuracy, parameter quantity, and FLOPs, the network structure proposed in this study demonstrated significantly superior performance compared to existing structures (excluding those previously searched by our research group). Moreover, it exhibited substantially higher discriminant accuracy than other lightweight network structures of the same type. The significance of the lightweight network lies primarily in the field of engineering, where it demonstrated superior performance while maintaining comparable accuracy, albeit its application is relatively limited.

Compared to the previously searched HSCNet by the research group (as presented in conference work), although the discrimination accuracy of the network proposed in this paper was slightly lower, it exhibited a significantly reduced number of FLOPs and parameters, approximately one-fourth that of HSCNet. Moreover, its architectural design is more conducive to efficient GPU reasoning and hardware platform utilization, rendering it particularly suitable for scenarios demanding high-speed discrimination capabilities.

The AGE-SPOS presented in this paper incorporates both the discrimination accuracy and inference speed of the network. When the MAE reached 1.80 years, its FLOPs was below 1B (with an input feature map size of 384 × 384). In terms of MAE/FLOPs/parameter count, AGE-SPOS significantly outperformed the existing network architectures and surpassed other lightweight networks of the same type in these three metrics. Compared to the HSCNet proposed by the research group, AGE-SPOS exhibited superior advantages in terms of both speed and network size. The present research further investigates the impact of the SE module on the accuracy and concludes that the SE module is not conducive to achieving accurate results.

To further explore the characteristics of the dental X-ray dataset, we developed the Cumulative Score (CS) as a method of measurement. CS is defined as:(8)$$CS(j)=\frac{N_{e}\leq j}{N}\times 100\%$$
where $N_{e}\leq j$ is the number of test images on which the age estimation makes an absolute error no higher than j year. When the error tolerance was set to a low value, significant disparities emerged among different models. Our model excelled compared to others, especially when the age error tolerance was set below 2 years. For instance, when the error tolerance was at 2 years, our model achieved a score of 0.80. In contrast, when the error tolerance was decreased to 1 year, our model still outperformed others with a score exceeding 70%, while other models struggled to reach this threshold. Additionally, it is worth noting that as the age error tolerance increased, the scores of all models started to converge, highlighting that the model’s superiority is most apparent in lower error tolerance ranges.

The performance of each age group was analyzed, as shown in Figure 9, which shows that the MAE in the 0–30 age range performed well in predicting within 2 years. The performance of each age group was analyzed, as depicted in Figure 9, revealing a favorable estimation effect within the age range of 0–30 years (MAE falls within 2 years). Notably, the most accurate predictions were achieved for individuals aged between 0 and 10 years old, with an MAE of approximately 1 year. The deep learning approach for age estimation has demonstrated superior performance compared to traditional methods during adolescence, and it also exhibited remarkable accuracy in estimating ages during adulthood. Although there were slight deviations in the results regarding the estimation of dental knowledge with respect to aging, such as inconsistent mean absolute error (MAE) values across different age ranges, it is important to note that these MAE values were derived solely from our limited dataset. In order to ensure accuracy, future research should encompass a broader range of datasets and conduct additional experiments. It is crucial to acknowledge that the current findings heavily rely on the available datasets.

Furthermore, we generated a heat-map visualization of the OPGs, as depicted in Figure 2. Upon careful examination of the heat maps, it became evident that the network effectively discerned distinct regions within the teeth. Notably, during adulthood age prediction, crucial factors predominantly resided in specific dental components such as the root area. The molar, upper, and lower jaw regions emerged as areas for identification, a fact that has eluded effective exploration through conventional methodologies.

## 5. Conclusions

In the article, we employed NAS to acquire a model that achieves the current optimal mean absolute error of 1.64. In comparison with Inception-v4, which has a computational complexity of 20.46 billion floating-point operations per second (FLOPs), our AGENet-Large module demonstrates a remarkable reduction in FLOPs by a factor of 2.7. Furthermore, we have successfully developed a series of lightweight models that can achieve an impressive MAE of 1.80 years while utilizing less than 1 billion FLOPs.

## Figures and Tables

**Figure 1 bioengineering-11-00674-f001:**
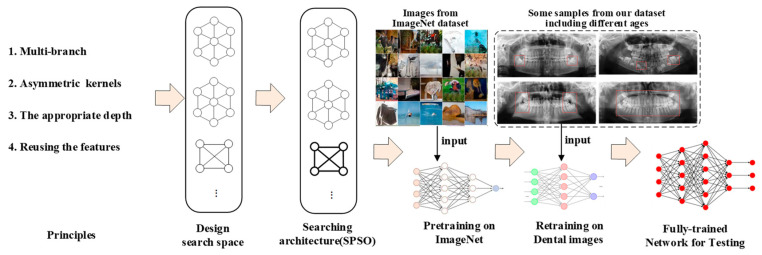
A searching framework for DNN architecture with X-ray image images. The framework consists of four main steps: (1) designing the search space, (2) exploring novel architectures, (3) pre-training the model on the ImageNet dataset, and (4) fine-tuning the pre-trained OPG datasets. The four proposed design rules were utilized to effectively design a search space for the purpose of obtaining an architecture that is capable of achieving optimal performance. Subsequently, the network underwent pre-training on the ImageNet dataset. Finally, the DNN architecture was retrained using the comprehensive training dataset of OPGs, leveraging the weights and parameters obtained from pre-training.

**Figure 2 bioengineering-11-00674-f002:**
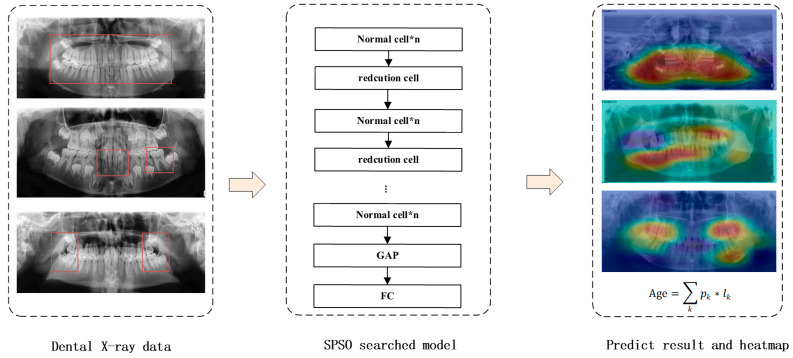
Age estimation based on the new dental X-ray images. By inputting the dental X-ray dataset into the DNN architecture, we were able to make accurate age estimations; the sum in the rightmost block is the average age for all images.

**Figure 3 bioengineering-11-00674-f003:**
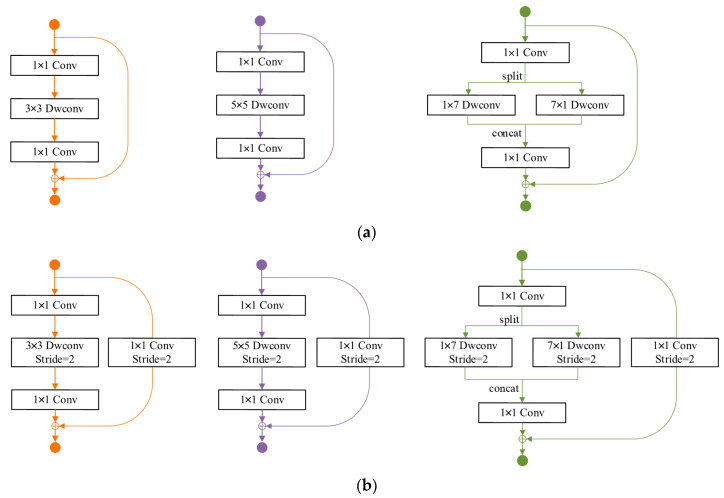
The configuration of this parallel asymmetrical convolutional unit. (**a**) The step is l for the search space; (**b**) the step is 2 for the search space.

**Figure 4 bioengineering-11-00674-f004:**
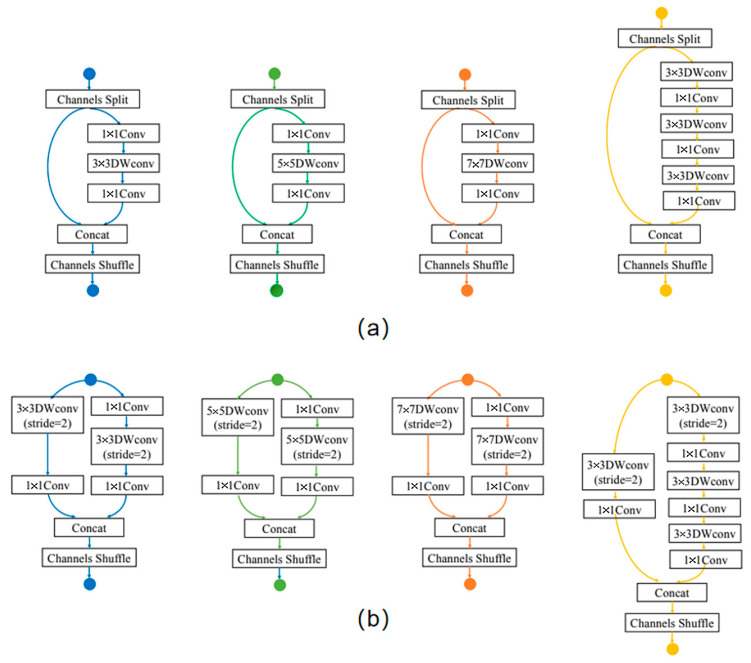
The search space under the SPOS framework. (**a**) Choice blocks with stride = l; (**b**) choice blocks with stride = 2.

**Figure 5 bioengineering-11-00674-f005:**
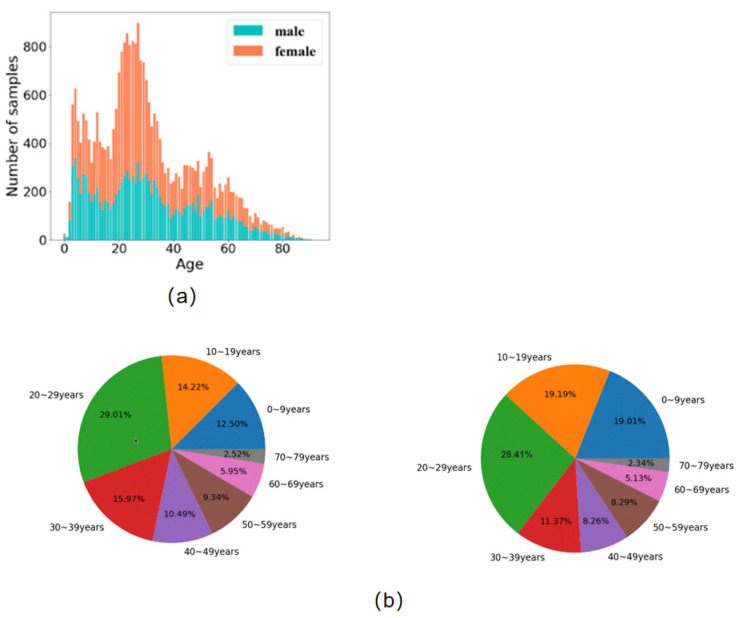
Age distribution of our age dataset. (**a**) Distribution of age; (**b**) training dataset for DNN (**left**) and test dataset for DNN (**right**).

**Figure 6 bioengineering-11-00674-f006:**
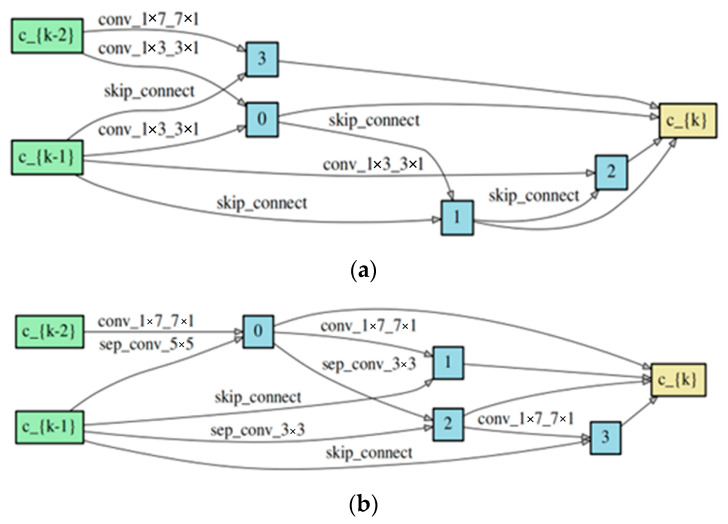
The typical cell and the reduction cell in the network. (**a**) The normal cell searched based on the input image; (**b**) the reduction cell searched based on the input images.

**Figure 7 bioengineering-11-00674-f007:**
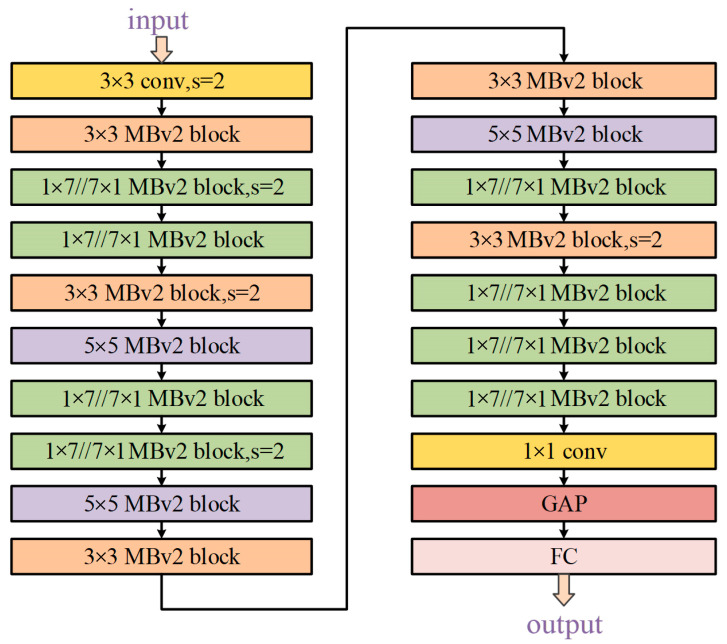
The architecture of AGE-SPOS with our searching method.

**Figure 8 bioengineering-11-00674-f008:**
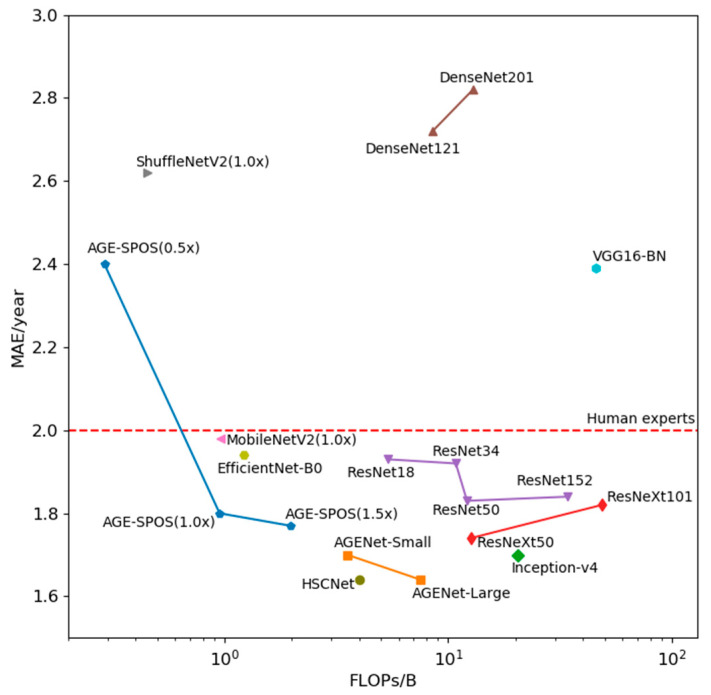
The accuracy and FLOPs for different DNN networks.

**Figure 9 bioengineering-11-00674-f009:**
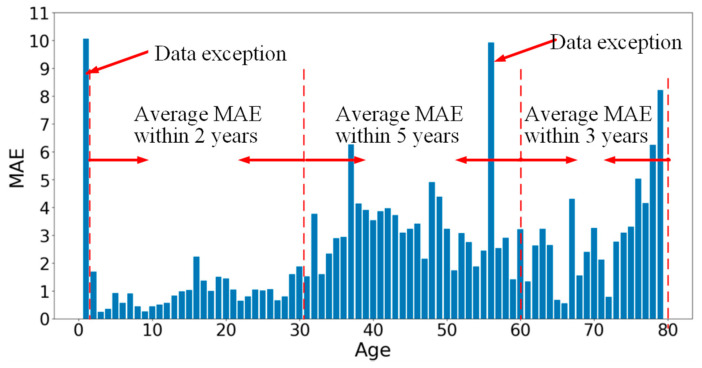
The MAE of different ages.

**Table 1 bioengineering-11-00674-t001:** The accuracy of AGE-SPOS with our searching method.

Model	FLOPs/s × 10^−9^	Params/s × 10^−6^	MAE/Year
Human experts	-	-	>2
ResNeXt50	12.59	23.14	1.74
Inception-v4	20.46	41.27	1.70
MobileNetV2 (1.0×)	0.96	2.33	1.98
EfficientNet-B0	1.22	4.11	1.94
AGENet-Small	3.54	7.53	1.70
AGENet-Large	7.49	16.89	1.64
HSCNet	3.99	7.78	1.64
AGE-SPOS (0.5×)	0.29	0.72	2.40
AGE-SPOS (1.0×)	0.95	2.17	1.80
AGE-SPOS (1.5×)	1.97	4.44	1.77

**Table 2 bioengineering-11-00674-t002:** The effectiveness of the SE module for different AGE-SPOS configurations.

DNN Architecture	MAE/Year	
	Without SE Module	With SE Module
AGE-SPOS (0.5×)	2.40	2.61 (+0.21)
AGE-SPOS (1.0×)	1.80	1.80 (−0.00)
AGE-SPOS (1.5×)	1.77	1.75 (−0.02)

## Data Availability

No new data were created or analyzed in this study. Data sharing is not applicable to this article.

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
