# Peer review of "Leverage Effective Deep Learning Searching Method for Forensic Age Estimation"

_bioengineering, 2024, doi:10.3390/bioengineering11070674_

Round 1

Reviewer 1 Report

Comments and Suggestions for Authors

In this manuscript, the authors compared multiple deep-learning methods for dental age estimation. The proposed approaches, AGENet and AGE-SPOS, outperformed in both accuracy and efficiency. Their findings could also inspire future research. In general, this manuscript reaches the required quality. 

Reviewer 2 Report

Comments and Suggestions for Authors

This manuscript describes an attempt to find an efficient DNN architecture for human age estimation from dental orthopantomograms (OPG). The PC-DARTS framework [32] is used to carry out an experimental search for the DNN structure given annotated images stored in a database comprising almost 30 thousand X-rays images. 

Consideration of the problem of the age estimation with the use of DNNs driven by OPG is motivated by important biometric applications and has attracted researchers attention for the last years. Optimizing the network architecture for the task seems to be an original contribution. What raises the reviewer concerns is the apparent lack of the presentation clarity and English text flaws. Examples are listed below.

32-34: The sentences are incomplete.

79-92: The cited references, related to segmentation of cardiac images, classification of cardiac rhythms and analysis of fundus images are not directly related to the topic of DNN-based age estimation from dental radiograms. A review of research work on this topic is available e.g. in  Mohamed et all. „Dental Age Estimation Using Deep Learning: A Comparative Survey”, Computation 2023, 11,18. This, and other papers are not considered as an expected part of state-of-the-art analysis of this submission. In this context, the statement in lines 114-115 does not seem convincing.

127: One would not know what is „the age probability distribution for every image”. How does this relate to the real subject age extracted from the ID data (line 20)?

130: What values the labels take? What is the range of summation on k?

133: „character DNN”?

172: The OPG acronym is reused in a different meaning.

183:  Graphical symbols in Figure 2 need explanation, especially those denoted by „Design search space” and „Searching architecture (SPSO)” which differ by 4 thick lines. The ImageNet dataset comprises RGB image unlike the OPG radiograms. How is that accounted for?

185: What does the age evaluated by the sum in the rightmost block of Figure 3 describe?

189-190: High efficiency of DARTS in memory utilization is in contradiction to their inefficiency in this matter, stated in line 203.

240: Captions a) and b) to Figure 4 should not have the form of sentences.

241: What are the „new OPGs”?

244: Which Nature journal is referred to?

253, 258: The „exp” symbol has already been used in expressions (3) and (4) to denote an exponential function.

257 and further on: Are F_m or FLOPs the measures of model computational complexity? What is a unit of it? FLOPS (the number of floating point operation per second) again?

 277: Figure 5 subcaptions (a) and (b) do not sound right.

278: Is Figure 5 show the age distribution really?

281: Please consider rephrasing the beginning of this paragraph, e.g.: „We have collected a substantial number of OPGs,” 

285-286: The sentence is badly formulated. First, images, not ages, are supposed to be used for training. Second, it „total of spanning ages(?)” are used, then what is the „remaining set”?

288-289: In what sense downsampling of the radiograms may increase the feature resolution? The ARP acronym has not been defined.

301-302: Please explain the meaning of label smoothing or give to a reference.  

312: Please use 1/N instead of the strange, calculator-like division symbol.

318: We initialized the channel to 36 —> We initialized the channel count to 36

341: The entries of Table 1 second column are values of computational complexity expressed in billions of floating-point operations per second? If so, the table captions should be reformatted accordingly. Similarly, the third column is supposedly the number of parameters in millions (or multiplied by 10^-6), and the rightmost is the mean absolute error in years?

357-358: Would this recommendation be valid for other OPG image datasets, different to the one collected by the authors?

366: The SE acronym has not been defined (squeeze-excitation?).

376-377: A verb is missing in the sentence.

385-386: Repeated sentence.

391-396, 412-413: Reference to the mentioned authors conference paper is missing.

405-407: What structures are referred to actually?

424: Which chapter?

429: The inequality in the numerator of (8) should be replaced by N_e. 

Comments on the Quality of English Language

Please see the above comments and suggestions for authors.

Reviewer 3 Report

Comments and Suggestions for Authors

The current manuscript requires some changes prior to acceptance, please address the points listed below.

Punctuation and commas should be corrected, many empty spaces are missing.​ Please invest more time in sentence formulations, etc.

DNN and come other abbreviations are introduced multiple times in the text: once is enough.

For all the figures, the captions should be extended to become understandable also without reading the main text.​ This is important!

The authors mention in the discussion a conference paper, supposedly on a similar subject.​ They should clarify to the reader what exactly are the essential differences of this apparently original submission as compared to those conference proceedings.

Interestingly, somewhat similar algorithms of categorization and model selection exist in the image analysis of single-particle-tracking recordings of diffusing particles. The authors can mention in the revised version some machine-learning and ​some Bayesian methods developed ​recently in Refs. [DOI 10.1088/1367-2630/ab6065] and [https://doi.org/10.1039/C8CP04043E].​ This is important for providing a wider coverage of the subject and of various methods.

What about incomplete data? How to compute the confidence of the proposed algorithms for such incomplete or​ for e.g. noisy data?

Conclusions should be extended substantially. In addition to dental-image analysis, what are other possible input data (bone​ fragments, ​ only parts of bodies (nails, hairs), etc.).   The text uses a lot of acronyms: provide a complete table of all the abbreviations prior to the bibliography, and may be also reduce their number by say 1/3.   ​Bibliography should be formatted with more care: all references should appear complete and in the same format.

Comments on the Quality of English Language

see the main report please
